**Data Availability Statement:** All relevant data are within the manuscript.

**Funding:** Sources of Funding: This study was supported by a research grant (IN-US-259-4294)

# Abacavir antiretroviral therapy and indices of subclinical vascular disease in persons with HIV

**Claudia A. Martinez**[1]*, **Rishi Rikhi**[2], **Mollie S. Pester**[3,4], **Meela Parker**[3], **Alex Gonzalez**[3], **Michaela Larson**[5], **Jennifer Chavez**[5], **Armando Mendez**[6], **Jeffrey K. Raines**[7], **Michael A. Kolber**[8], **Ivonne H. Schulman**[9], **Maria L. Alcaide**[8], **Barry E. Hurwitz**[3,4,6]

1 Division of Cardiology, University of Miami Miller School of Medicine, Miami, Florida, United States of America, 2 Division of Cardiology, Wake Forest University School of Medicine, Winston-Salem, North Carolina, United States of America, 3 Behavioral Medicine Research Center, University of Miami, Miami, Florida, United States of America, 4 Department of Psychology, University of Miami, Coral Gables, Florida, United States of America, 5 Department of Public Health Science, University of Miami Miller School of Medicine, Miami, Florida, United States of America, 6 Division of Endocrinology, Diabetes and Metabolism, Miller School of Medicine, University of Miami, Miami, Florida, United States of America, 7 Department of Surgery, University of Miami Miller School of Medicine, Miami, Florida, United States of America, 8 Division of Infectious Disease, Department of Medicine, University of Miami Miller School of Medicine, Miami, Florida, United States of America, 9 Katz Family Division of Nephrology & Hypertension, University of Miami Miller School of Medicine, Miami, Florida, United States of America

* Cmartinez5@med.miami.edu

## Abstract

### Objective

Indices of cardiovascular disease (CVD) risk, vascular endothelial dilation, arterial stiffness and endothelial repair were examined in persons with HIV (PWH) on an antiretroviral therapy (ART) that included abacavir (ABC+) in comparison with PWH on ART without abacavir (ABC-), and with HIV seronegative (HIV-) individuals.

### Approach

The 115 participants (63% men), aged 30–50 years, did not have CVD, metabolic, endocrine, or chronic renal conditions. PWH were on stable ART for six-months or more. Vascular assessments included flow-mediated dilation (FMD), aortic, radial and femoral arterial stiffness (cAIx, crPWV, cfPWV), and thigh and calf arterial compliance (Vmax50). Endothelial repair was indexed by endothelial progenitor cell colony forming units (EPC-CFU). Traditional CVD risk measures included blood pressure, central adiposity, lipids, insulin resistance (HOMA-IR), CRP and ASCVD score. Analyses controlled for demographics (age, sex, education), medications (antihypertensive, statin/fibrate, antipsychotic), and substance abuse (ASSIST).

### Results

No group differences were observed in central adiposity, HOMA-IR, CRP, or ASCVD risk score. However, the ABC- group displayed greater dyslipidemia. The ABC+ group displayed no difference on FMD, cAIx, cfPWV or calf Vmax50 compared with other groups. When

awarded to CAM and BEH from Gilead Sciences, Inc, and by the Miami CFAR (P30AI073961). IHS currently works at the National Institutes of Health. She contributed to this manuscript in her personal capacity. The opinions expressed in this article are the author's own and do not reflect the view of the National Institutes of Health, the Department of Health and Human Services, or the United States government. The other authors have no disclosures. The funders had no role in study design, data collection and analysis, decision to publish, or preparation of the manuscript.

**Competing interests:** The authors have declared that no competing interests exist.

CD4 count and viral load were controlled, no additional differences between the ABC+ and ABC- groups emerged. Analyses of crPWV and thigh Vmax50 suggested supported by a trend toward lower EPC-CFU in the HIV+ groups than the HIV- group.

## Conclusions

Findings indicate that ABC treatment of 30–50 year-old PWH on stable ART is not likely to contribute in a robust way to higher CVD risk.

## Introduction

Persons living with HIV (PWH) have elevated subclinical cardiometabolic complications and have a 1.5-2-fold higher incidence of cardiovascular disease (CVD) compared with seronegative counterparts [1, 2]. The primary cardiometabolic complications include central adiposity, dyslipidemia, insulin resistance, dysglycemia, as well as hypertension [3–5]. HIV infection promotes immune activation and increases oxidative stress causing vascular endothelial dysfunction and decreased endothelium repair leading to premature vascular dysfunction and subclinical CVD pathophysiology [6–8]. Subclinical vascular pathogenesis may be detected using measures of vascular endothelial dysfunction, indexed by impaired flow-mediated dilation (FMD) measured by ultrasound imaging of the brachial artery [9]. In addition, elevated arterial stiffness (i.e., reduced arterial compliance), estimated using peripheral artery tonometry and other cuff-based oscillometric methods, are predictively associated with increased CVD pathophysiology and events [10–13]. Similarly, the presence of vascular disease, as indexed by lower numbers of endothelial progenitor cells colony forming units (EPC-CFU), is a marker of impaired endothelial repair and CVD morbidity and mortality [14, 15].

Despite the advent of combined antiretroviral therapy (ART), which has substantially improved the survival of PWH, certain ART medications may be associated with elevated CVD risk [16–18]. The present study arises from research in the past decade that has revealed an association between the nucleoside reverse transcriptase inhibitor abacavir (ABC) and CVD events in PWH [19, 20]. However, this association is not without controversy. A recent review of this literature indicates that the association of ABC treatment with CVD events in PWH is more nuanced and may depend on the influence of ART treatment history and duration and the confluence of sociodemographic and CVD risk factors, comorbid conditions and treatments, and substance abuse history [21]. A few previous studies of PWH using vascular endothelial function and arterial stiffness measures have attempted to clarify the association of ABC treatment with subclinical CVD risk [22–27], but the association of the use of ABC with CVD event outcomes, have been inconsistent, some showing evidence of vascular endothelial function and arterial stiffness and others not. One problem in nonrandomized cohort studies of ABC treatment is that ABC was preferentially prescribed to those at higher underlying risk of CVD, where clinicians had avoided the use of other nucleoside reverse transcriptase inhibitors (NRTIs) that were known to have adverse lipid effects resulting in potential channeling bias [19]. Thus, to minimize channeling and selection bias, the present study was designed with strict eligibility criteria and statistical control for potentially confounding demographic, cardiometabolic comorbidities, other medication use and substance abuse.

The study objective was to examine indices of vascular function and repair in PWH on stable ART regimens (no change in ART regimen in the last 6 months before enrollment) with abacavir treatment (ABC+) in comparison with PWH on stable ART regimens that do not

include abacavir (ABC-), and with HIV seronegative (HIV-) individuals. We hypothesized that ABC-treated PWH on would manifest greater vascular dysfunction when compared to the other groups. The study aims were to evaluate whether group differences were observed: 1) on measures of endothelial-dependent vasodilation, arterial stiffness, arterial compliance, and endothelial repair; and 2) when controlling for a) demographic variables (sex, age, education); b) prescribed medications for comorbid conditions; and c) extent of substance abuse.

## Material and methods

### Participants

The study was reviewed and approved by the Human Subjects Research Committees of the University of Miami. The institutional review board approval # 20161218 and all participants provided written informed consent. Participants were recruited from the outpatient adult HIV Clinics of the University of Miami and Jackson Memorial Hospital complex, and the study was conducted in collaboration with the Miami CFAR during the period of November 2017 until January 2020. Study eligible PWH and HIV- participants were 30 to 50 years of age. Although persons with hypertension and/or dyslipidemia were eligible for the study, persons were excluded with any other diagnosed cardiovascular, metabolic, or endocrine condition. Persons with end stage renal disease or chronic renal disease stage 3 or greater, morbid obesity (body mass index (BMI) > 40 kg/m$^2$), or seizure disorder were also excluded. HIV serostatus of PWH was confirmed via patient medical records. In addition, PWH were eligible if they had been on stable ART therapy without any change in the last 6 months prior to enrollment and confirmed by review of the medical records. Participants were excluded if they were treated with didanosine, an NRTI associated with increased risk of cardiovascular events [28]. In addition, women were excluded if pregnant, breast feeding, or postmenopausal, to minimize variability of endogenous sex hormones among participants and potential influence on cardiovascular function [29]. Similarly, persons on gender reaffirming hormone therapy in the previous year were also excluded. To minimize potential major sources of bias, the study recruitment procedure matched ABC- and HIV- participants to ABC+ participants by the sex, ethnicity/race and BMI of the ABC+ group by periodically monitoring group compositions and attempting to recruit and enroll based on the average group proportions. The vascular measures were conducted at the Behavioral Medicine Research Center of the University of Miami.

### Study protocol and procedures

The study consisted of a single assessment visit, wherein study eligibility was confirmed and study outcomes were obtained. Upon arrival at the laboratory following a fast from midnight, demographic indices (age, sex, education and race/ethnicity), anthropometric measures (height, weight, BMI, and waist circumference), and relevant medical history (e.g., diagnosed conditions and prescribed medications). History of substance use was obtained via the Alcohol, Smoking and Substance Involvement Screening Test (ASSIST), which is a validated structured interview [30]. This instrument identifies lifetime and current use history of tobacco, alcohol, cannabis, cocaine, stimulants, inhalants, sedatives, hallucinogens, opioids, and other substances. A substance dependence and an abuse score were derived from these measures. For PWH, HIV diagnosis date, CD4 count, HIV-1 viral load, current ART regimen and treatment duration was obtained from medical records and confirmed via self-report when records were incomplete. After the collection of this information, the cardiovascular assessment was undertaken; following a 10-min rest period, resting blood pressure was obtained and the FMD test was performed, followed by the assessments of arterial stiffness and compliance. Then, a

venous blood sample was drawn and subsequently assayed for fasting glucose and insulin, C-reactive protein (CRP), lipid profile, and EPC-CFU.

**Endothelial-dependent vasodilatory function.** The FMD measure was obtained from the brachial artery reactive hyperemia test, while participants lay supine [31]. B-mode ultrasound scan of the right brachial artery was obtained in the longitudinal scanning plane about 5 cm above the elbow for clear visualization of the posterior wall intima-lumen and the anterior wall media-adventitial interfaces. An 11 MHz linear array transducer (Sonos 5500; Philips) was held in-place by stereotactic clamp. To optimize image quality, fine adjustments were made by means of micromanipulators attached to the clamp mount permitting horizontal and vertical axis rotation. Arterial flow was manipulated by a pneumatic cuff affixed to the forearm distal to the arterial segment being imaged. Cuff inflation (240 mm Hg) and deflation was performed by an electronic controller. Following 15 min of supine rest, recordings were taken for 1 min at pre-cuff inflation baseline, 5 min during inflation, and 5 min after deflation. Following 5 min of supine rest, a second FMD test was performed. The image analysis obtained about 150-paired lumen measurements along a 30 mm arterial wall length at 30 ms intervals. Hence, arterial diameter was provided through the cardiac cycle, permitting reliable artifact rejection. The percent change in arterial diameter during diastole at peak response was quantified for the two FMD tests and the mean obtained. Measurement reproducibility is very high for within-session (systolic: $r = .999$, CV = 0.8%; diastolic: $r = .997$, CV = 1.6%) and between-sessions (systolic: $r = .997$, CV = 1.4%; diastolic: $r = .991$, CV = 2.6%).

**Arterial stiffness.** Following the FMD tests and another 5 min supine rest period, arterial stiffness evaluation was performed by the same technician using applanation tonometry (SphygmoCor; AtCor Medical) as per consensus standards [32]. The central augmentation index (cAIx) and pulse wave velocity (PWV) are measures of arterial stiffness. The cAIx is a measure of wave reflection amplitude derived from aortic pressure wave analysis [33]. The pressure wave augmentation (AP) was calculated as the difference between the second (P2; caused by wave reflection) and the first systolic peak pressure (P1; caused by ventricular ejection). The cAIx was derived as the quotient of the AP and central pulse pressure (cPP). For this index, the pulse wave is recorded from the radial artery of the right arm using a high fidelity micromanometer (Millar Instruments). The software generates a model of the central pressure waveform by using a validated generalized transfer function. Using simultaneous ECG-R wave gating, the time delay of the pulse wave from the carotid to femoral and carotid to radial segments (i.e., cfPWV and crPWV, respectively) was obtained. The system computed the PWV as the ratio of the distance between the respective carotid to femoral and carotid to radial measurement sites with the corresponding time delay. The carotid-femoral and carotid-radial distances were estimated by subtracting the sternal notch to the carotid site distance from the distance from the sternal notch to the respective femoral and radial sites [10]. The mean of three consecutive readings were used for the cAIx and PWV measures; each reading consisted of at least 10–20 sequentially recorded waveforms. The system provided quality control of the recorded pressure waves comprising the variations in pulse height, diastole, and pulse length; acceptable variation was ≤7%. These methods have been reported to have good reproducibility (intra-observer variation: cAIx = 0.0±11%; cfPWV = -0.1 ±2.5 m/s; crPWV = 0.0 ±1.3 m/s) [34]. Notably, invasive PWV comparisons with noninvasive PWV measures demonstrate good agreement [35].

To further index arterial stiffness, an automated, computer-controlled air plethysmograph was used (Soteria Cardiac Platform, Soteria Medical, Miami, FL). This device consists of an air pump, calibration chamber (i.e., piston), high-resolution pressure transducer, and three standard blood pressure cuffs placed at the brachial, thigh, and calf levels [36]. The brachial level cuff was placed on a single upper extremity to measure systemic blood pressure (systolic,

diastolic, and mean pressures). Segmental limb volume changes were measured independently at the thigh and calf levels during the entire cardiac cycle. The segmental volume change in the thigh and calf were initially obtained at a pressure of 50 mm Hg. The cuff pressure automatically was increased by the device in 10 mm Hg increments, and segmental volume change measurements were obtained at each pressure until the segmental volume change was maximized. At each cuff pressure during early diastole, a known calibration volume of 0.65 mL was delivered via the piston rapidly expanding the closed cuff system for the thigh and calf regions independently. The maximum volume (Vmax) change during this sequence was normalized to 50 mm Hg and calculated as follows: Vmax50 = maximum $\Delta V^*50$/pulse pressure. Thus, thigh and calf Vmax50 arterial compliance measures were obtained, indicating volume (ml) for each 50 mm Hg of pulse pressure per region.

**Fasting glucose, insulin, and lipid profile.** A total of 5 mL venous sample was obtained and assays performed to obtain fasting plasma glucose (FPG) and insulin (FPI) [31]. The homeostasis model was used to derive an insulin resistance estimate (HOMA-IR) [37]. In addition, total cholesterol (TC), triglycerides (TG), HDL-cholesterol and high-sensitivity C-reactive protein (CRP) were measured on a Roche 6000 Auto-Analyzer (Roche Diagnostics). Automated chemistry and immunoassays exhibited interassay coefficients of variation consistently less than 5%. LDL-cholesterol was calculated by the Friedewald equation [38].

**Endothelial repair.** From 45 mL of intravenous blood, measures of EPC-CFU were obtained by isolating early EPCs using Ficoll Paque and seeding 5 million cells on 6-well Fibronectin-coated dishes (BD Biosciences) in CFU-Hill medium (Stem Cell Technologies, cat#05900). Non-adherent cells were collected 48 hours later, and 1 million cells seeded on 24 well Fibronectin-coated dishes. On day 5, EPC-CFUs were counted in 5 wells. The mean EPC-CFU was used in analyses as previously described [14, 39].

**Assessment of cardiovascular risk.** Computation of atherosclerotic cardiovascular disease risk (ASCVD) was performed using the ASCVD estimator to derive 10-year risk [40]. The ASCVD algorithm considers sex, age, race, and personal history of diabetes, smoking status and hypertension treatment, in addition to TC, HDL, and SBP levels [41].

## Statistical analysis

Data screening using SPSS v26.0 (IBM Corp., Armonk, NY) included calculating descriptive statistics, intercorrelations, variable distributions, and evaluating normality via Kolmogorov-Smirnov. Transformation of variables were performed for continuous measures departing from normality (i.e., HOMA-IR, FPG, FPI, TG, HDL, TG/HDL, cPP, and EPC-CFU). Complete data for all variables was collected except for FPG, FPI, CRP, and lipid profile (2.6% missing), EPC-CFU (6.1% missing). Multiple imputation was used to estimate these data, assumed to be missing at random [42]. In addition, data were missing for cAIx, cfPWV, crPWV and cPP (19.1%), and thigh and calf Vmax50 (7.0%) measures due to technical error. Group (ABC+ vs. ABC- vs. HIV-) differences on outcome measures were examined using analysis of variance (ANOVA) for Aim 1, and analysis of covariance (ANCOVA) for Aims 2a-c. For significant group effects ($\alpha$ = .05), post-hoc pairwise comparisons were conducted using Fisher's LSD test.

## Results

### Participant characteristics

Demographic and other cohort characteristics for study groups are described in Table 1. Of the 160 screened participants, data were collected from 115 study eligible participants, which included 28 HIV+ ABC treated, 39 HIV+ non-ABC treated, and 48 HIV- persons. As seen in

**Table 1. Sociodemographic, clinical, and HIV-Related characteristics of participants by study group[a].**

| Measures | | | ABC+ (1) | ABC- (2) | HIV- (3) | Comparison[b] |
|---|---|---|---|---|---|---|
| | | | Mean SE | Mean SE | Mean SE | |
| **Demographics** | | | | | | |
| Age | | (yr) | 42.2 ± 1.1 | 42.2 ± 0.8 | 38.2 ± 0.8 | ** 1 = 2>3 |
| Sex | | (%) | | | | ns |
| | Men | | 64.5 | 66.7 | 62.5 | |
| Ethnicity/Race | | (%) | | | | ns |
| | Black | | 38.7 | 56.3 | 56.3 | |
| | Hispanic | | 48.4 | 29.2 | 33.3 | |
| | non-Hispanic white | | 6.5 | 8.3 | 8.3 | |
| Education | | (yr) | 11.8 ± 0.6 | 12.8 ± 0.5 | 14.8 ± 0.5 | ***1 = 2<3 |
| **Clinical Characteristics** | | | | | | |
| BMI: | overweight | (%) | 45.2 | 35.4 | 50.0 | ns |
| | obesity | (%) | 38.7 | 29.2 | 27.2 | ns |
| [c]TG: | borderline high | (%) | 7.1 | 17.9 | 0.0 | * 2>1 = 3 |
| | high | (%) | 10.7 | 20.5 | 10.4 | ns |
| | median (IQR) | (mg/dl) | 102.0 (74.0) | 121.0 (66.0) | 78.0 (38.8) | |
| [d]TC: | borderline high | (%) | 25.8 | 33.3 | 18.8 | ns |
| | high | (%) | 9.7 | 12.5 | 4.2 | ns |
| | median (IQR) | (mg/dl) | 174.5 (46.8) | 202.0 (53.0) | 173.5 (37.5) | |
| ASCVD 10 yr risk | | (%) | 2.7 ± 0.5 | 3.0 ± 0.3 | 2.4 ± 0.4 | ns |
| Dependence score | | (U) | 17.0 ± 3.6 | 23.4 ± 2.9 | 14.7 ± 2.9 | [t] 2>3 = 1 |
| Drug dependence | | (%) | 32.3 | 37.5 | 25.0 | ns |
| Abuse score | | (U) | 11.6 ± 2.4 | 17.0 ± 1.9 | 10.7 ± 1.9 | [t] 2>3 = 1 |
| Drug abuse | | (%) | 12.9 | 22.9 | 10.4 | ns |
| Anti-hypertensive med | | (%) | 38.7 | 6.3 | 2.1 | ***1>2 = 3 |
| Statin/Fibrate med | | (%) | 19.4 | 2.1 | 2.1 | ** 1>2 = 3 |
| Anti-psychotic med | | (%) | 16.1 | 8.3 | 4.2 | ns |
| Thyroid med | | (%) | 3.2 | 2.1 | 2.1 | ns |
| **HIV-related Characteristics** | | | | | | |
| Known HIV duration | | (yr) | 14.8 ± 1.6 | 13.2 ± 1.4 | | ns |
| Current ART duration | | (yr) | 3.1 ± 0.4 | 2.7 ± 0.3 | | ns |
| CD4 T cell count | | (cells/mm$^3$) | 614.9 ± 60.0 | 511.6 ± 68.5 | | ns |
| HIV-1 viral load | | (log copies/ml) | 1.6 ± 0.1 | 1.6 ± 0.8 | | ns |
| Undetected viral load | | (%) | 82.1 | 69.2 | | ns |

abbreviations: BMI = body mass index; TG = triglycerides; TC = total cholesterol; ASCVD = atherosclerotic cardiovascular disease risk; ART = antiretroviral therapy.

[a]ABC+ n = 28, ABC- n = 39, HIV- n = 48.

[b]$\chi^2$, t-test, or ANOVA of group differences;

*p < .05;

**p < .01;

***p < .001;

[t]p < .10; ns = not significant.

[c]TG borderline high is 150–199 mg/dl; high is 200–499 mg/dl; very high is >500 mg/dl.

[d]TC borderline high is 200–239 mg/dl; high is >240 mg/dl.

Table 1, no significant differences were found in group composition based on sex or ethnicity/race. However, the HIV- group compared with the other groups was about 4 years younger and had about 2 years more education on average. Of note, no significant group difference in BMI was observed, nor were differences in the proportion of the groups who were overweight or obese observed, with approximately 74.6% of participants classified as such. Groups were comparable in the proportion of participants who were classified with elevated TG and TC, although relative to the HIV- group, the HIV+ groups evidenced more persons who were classified with mildly increased TG. However, there were no significant difference among groups in ASCVD 10-year risk score. As seen in Table 1, there was a trend towards greater drug dependence and abuse scores on the ASSIST survey in the ABC- group than other groups; however, no group differences emerged in the proportion of persons classified using the clinical criteria with drug dependency or abuse. In sum, of the study participants, 34.1% were classified with drug dependence and 15.3% with drug abuse. Regarding prescribed medication use, relative to the other groups, the ABC+ group had significantly greater use of anti-hypertensive, statin and fibrate medications, but no group differences were found in the use of anti-psychotic or thyroid medications. As can be seen in Table 1, the HIV-related characteristics of the ABC+ and ABC- groups were comparable in known HIV duration, years on current ART regimen, CD4 count, HIV viral load, and percent of participants with undetectable viral load (≤20 copies/ml). Table 2 shows the ART medication regimens for the ABC+ and ABC- groups, which included nucleoside reverse transcriptase inhibitors (NRTI), non-nucleoside reverse transcriptase inhibitors (NNRTI), integrase inhibitors (II), and protease inhibitors (PI). No participants were treated with fusion inhibitors (FI) or entry inhibitors (EI). For the ABC+ group, in addition to ABC, a majority (93%) were treated with IIs. In contrast, most of the ABC- group, were treated with an NRTI/II boosted, NRTI/PI boosted or NRTI/II regimens.

## Subclinical cardiometabolic and vascular outcomes by group (Aim 1)

Table 3 displays differences of subclinical cardiometabolic measures among groups. The analysis indicated that there was no significant difference between the groups on BMI nor in central adiposity indexed by waist circumference. Insulin resistance, assessed by HOMA-IR, trended toward higher levels in the ABC+ than the other groups, a finding which is supported by a trend towards greater FPI in the ABC+ group. There were no significant group differences in

**Table 2. ART medication regimens for the ABC+ and ABC- groups.**

| | ABC+ (1) | | ABC- (2) | |
|---|---|---|---|---|
| Medication Regimen | n | % | n | % |
| NRTI exclusively | 0 | 0 | 3 | 8 |
| PI exclusively | 0 | 0 | 1 | 3 |
| NRTI/NNRTI | 0 | 0 | 1 | 3 |
| NRTI/II | 26 | 93 | 7 | 18 |
| NNRTI/II | 0 | 0 | 1 | 3 |
| PI boosted | 0 | 0 | 2 | 2 |
| NRTI/II/PI | 1 | 4 | 0 | 0 |
| NRTI/II boosted | 0 | 0 | 17 | 44 |
| NRTI/PI boosted | 1 | 4 | 7 | 18 |

abbreviations: NRTI = nucleoside reverse transcriptase inhibitors, NNRTI = non- nucleoside reverse transcriptase inhibitors, II = integrase inhibitors, PI = protease inhibitors.

**Table 3. Subclinical cardiometabolic measures comparing study groups.**

| Measures | | ABC+ (1) | ABC- (2) | HIV- (3) | Comparison[a] |
|---|---|---|---|---|---|
| | | Mean SE | Mean SE | Mean SE | |
| **CVD Risk** | | | | | |
| BMI | $(kg/m^2)$ | 29.0 ± 0.8 | 27.0 ± 0.8 | 28.0 ± 0.6 | ns |
| Waist circumference | (cm) | 98.7 ± 3.3 | 93.8 ± 2.4 | 92.9 ± 2.1 | ns |
| HOMA-IR[c] | (mass units) | 3.4 ± 0.5 | 2.3 ± 0.2 | 2.3 ± 0.2 | [t] 1>2 = 3 |
| FPG[c] | (mg/dl) | 94.6 ± 3.0 | 89.9 ± 1.3 | 89.2 ± 1.5 | ns |
| FPI[c] | (μU/ml) | 13.9 ± 1.8 | 10.4 ± 0.9 | 10.1 ± 0.8 | [t] 1>2 = 3 |
| CRP | (mg/L) | 3.1 ± 0.6 | 2.6 ± 0.4 | 2.6 ± 0.4 | ns |
| TG[c] | (mg/dl) | 119.4 ± 13.1 | 142.2 ± 10.3 | 106.4 ± 13.4 | **2>3 |
| TC | (mg/dl) | 180.2 ± 6.7 | 201.9 ± 5.4 | 178.2 ± 4.9 | **2>3 = 1 |
| LDL | (mg/dl) | 104.3 ± 6.3 | 120.7 ± 5.1 | 102.5 ± 3.9 | * 2>3 = 1 |
| HDL | (mg/dl) | 52.1 ± 2.6 | 52.5 ± 3.1 | 54.4 ± 1.9 | ns |
| TC/HDL | | 3.7 ± 0.2 | 4.2 ± 0.2 | 3.5 ± 0.2 | * 2>3 = 1 |
| TG/HDL[c] | | 2.6 ± 0.4 | 3.1 ± 0.3 | 2.3 ± 0.4 | **2>3 |
| SBP | (mm Hg) | 116.2 ± 2.6 | 115.6 ± 2.1 | 117.9 ± 2.5 | ns |
| MAP | (mm Hg) | 89.5 ± 2.0 | 87.5 ± 1.5 | 87.0 ± 1.6 | ns |
| DBP | (mm Hg) | 74.3 ± 1.6 | 72.1 ± 1.2 | 70.4 ± 1.3 | ns |
| Heart rate | (bpm) | 67.6 ± 1.2 | 65.1 ± 1.6 | 61.6 ± 1.3 | **1 = 2>3 |
| **Vascular Function** | | | | | |
| FMD | (%) | 7.0 ± 0.6 | 6.4 ± 0.4 | 7.9 ± 0.5 | [t] 1 = 2<3 |
| cAIx[d] | (%) | 21.9 ± 2.5 | 17.2 ± 2.6 | 16.9 ± 2.1 | ns |
| cfPWV[d] | (m/s) | 6.9 ± 0.3 | 6.6 ± 0.2 | 6.4 ± 0.1 | ns |
| crPWV[d] | (m/s) | 7.6 ± 0.3 | 8.2 ± 0.3 | 7.4 ± 0.1 | * 2>3 = 1 |
| cPP[cd] | (mm Hg) | 29.7 ± 1.3 | 29.1 ± 1.1 | 33.6 ± 1.6 | [t] 2 = 1<3 |
| Thigh Vmax50[e] | (ml) | 3.3 ± 0.2 | 3.2 ± 0.2 | 3.5 ± 0.2 | ns |
| Calf Vmax50[e] | (ml) | 1.9 ± 0.1 | 1.9 ± 0.1 | 2.0 ± 0.1 | ns |
| EPC-CFU[c] | (U) | 3.9 ± 1.5 | 4.6 ± 2.1 | 5.1 ± 0.6 | [t] 3>2 = 1 |

abbreviations: BMI = body mass index, HOMA-IR = homeostatic model assessment for insulin resistance, FPG = fasting plasma glucose, FPI = fasting plasma insulin, CRP = C reactive protein, TG = triglycerides, TC = total cholesterol, LDL = low density lipoprotein-cholesterol, HDL = high density lipoprotein-cholesterol, SBP = systolic blood pressure, MAP = mean arterial pressure, DBP = diastolic blood pressure, FMD = flow mediated dilation, cAIx = central augmentation index, cfPWV = carotid-femoral pulse wave velocity, crPWV = carotid-radial pulse wave velocity, cPP = central pulse pressure, Thigh Vmax50 = thigh arterial compliance, Calf Vmax50 = calf arterial compliance, EPC-CFU = endothelial progenitor cell-colony forming unit.

[a] means ± SE values displayed are from non-transformed variables.

[b] $\chi^2$ or ANOVA of group differences;

*p < .05;

**p < .01;

***p < .001;

[t] p < .10; ns = not significant.

[c] group comparison reflects analysis of transformed values.

[d] ABC+ n = 24, ABC- n = 26, HIV- n = 43.

[e] ABC+ n = 24, ABC- n = 37, HIV- n = 46.

CRP levels. The analyses of lipid profile showed that, although no group differences were observed in HDL levels, TG, TC and LDL levels were significantly greater in the ABC- group relative to counterparts; similar group differences were found in TC/HDL and TG/HDL ratios. The analyses of cardiovascular indices showed no group difference in blood pressure. Heart

rate level was greater in both the HIV+ groups than the HIV- group ($p < .01$). The analyses of the vascular function measures indicated a trend toward lower FMD in the HIV+ groups compared with other HIV- group. The measure of cPP was trended toward lower levels in the HIV+ groups compared with the HIV- group. However, of the measures of arterial stiffness, groups differences were found only in crPWV; significantly higher crPWV was observed in the ABC- group compared with the other groups ($p < .05$). Analysis of EPC-CFU levels showed a trend toward lower levels in the HIV+ groups than the HIV- group.

## Study outcomes controlling for demographic variables (Aim 2a)

Table 4 summarizes group differences in subclinical cardiometabolic and vascular measures when controlling for demographic variables (sex, age, education). All findings reported above remained except that group differences in HOMA-IR, which were trending, became significant ($p < .05$) and FPG became a trend, wherein the ABC+ group displayed higher levels than the other groups. Whereas group differences in lipid profile remained, differences among groups were no longer significant in FMD and EPC-CFU levels. The analyses of vascular function measures indicated that group differences in crPWV, which were significant, became trending, although the previously observed trend toward group differences in cPP became significant. In addition, the measure of thigh vascular compliance (Thigh Vmax50) became significant ($p < .05$), wherein lower compliance was observed in the ABC- group than the HIV- group.

## Study outcomes controlling for demographics and prescribed medication use (Aim 2b)

In Table 5, group differences are displayed on subclinical cardiometabolic and vascular measures among groups when prescription medication variables (antihypertensive, statin-fibrate, antipsychotic, thyroid) were added to the model. Analyses showed there was no longer any group differences on metabolic measures (HOMA-IR, FPI, FPG). All other previously observed significant group differences were maintained for TG, TC, LDL, TC/HDL, TG/HDL, heart rate, crPWV, cPP and thigh Vmax50.

## Study outcomes controlling for demographics, prescribed medications and substance abuse (Aim 2c)

Table 6 summarizes the analysis of group differences on outcomes when controlling for substance abuse score in addition to demographic and medication variables. Note, due to multi-collinearity with the substance abuse score, the substance dependence score was not used as a covariate in analyses. The analysis showed that group differences remained on TG, TC, LDL, TC/HDL, TG/HDL, heart rate, crPWV, cPP and thigh Vmax50, when controlling for these variables. When this analytic model was restricted to the two HIV+ groups, and the CD4 count and HIV-1 viral load measures were added into the model as covariates, the differences between ABC+ and ABC- groups did not differ from that described above.

## Discussion

The balance among biomechanisms influencing vascular damage and repair appear to be impaired in the context of HIV/AIDS [43–45]. In the setting of endothelial damage, EPCs are released from bone marrow, home to target areas, and incorporate and promote neovascularization and paracrine factors involved in vascular repair [46]. Hence, endothelial injury without a sufficient compensatory reparative response is thought to accelerate the development of

**Table 4. Subclinical cardiometabolic measures comparing study groups when controlling for demographic variables (sex, age, education).**

| Measures | | ABC+ (1) | ABC- (2) | HIV- (3) | Comparison[a] |
|---|---|---|---|---|---|
| | | Mean SE | Mean SE | Mean SE | |
| **CVD Risk** | | | | | |
| BMI | (kg/m$^2$) | 28.7 ± 0.9 | 27.1 ± 0.7 | 28.0 ± 0.7 | ns |
| Waist circumference | (cm) | 98.1 ± 3.0 | 93.4 ± 2.6 | 93.6 ± 2.4 | ns |
| HOMA-IR[c] | (mass units) | 3.5 ± 0.3 | 2.3 ± 0.3 | 2.3 ± 0.3 | * 1>2 = 3 |
| FPG[c] | (mg/dl) | 95.1 ± 2.2 | 89.8 ± 1.9 | 89.2 ± 1.7 | [t] 1>2 = 3 |
| FPI[c] | (μU/ml) | 14.1 ± 1.3 | 10.4 ± 1.1 | 10.0 ± 1.0 | [t] 1>2 = 3 |
| CRP | (mg/L) | 2.8 ± 0.6 | 2.5 ± 0.5 | 2.8 ± 0.5 | ns |
| TG[c] | (mg/dl) | 123.7 ± 15.1 | 140.4 ± 12.8 | 105.4 ± 11.9 | **2>3 |
| TC | (mg/dl) | 178.7 ± 6.7 | 200.8 ± 5.7 | 179.9 ± 5.2 | * 2>3 = 1 |
| LDL | (mg/dl) | 103.8 ± 5.9 | 121.2 ± 5.0 | 102.4 ± 4.6 | * 2>3 = 1 |
| HDL | (mg/dl) | 50.2 ± 3.0 | 51.4 ± 2.5 | 56.4 ± 2.3 | ns |
| TC/HDL | | 3.7 ± 0.3 | 4.2 ± 0.2 | 3.4 ± 0.2 | * 2>3 |
| TG/HDL[c] | | 2.8 ± 0.4 | 3.1 ± 0.4 | 2.2 ± 0.4 | **2 = 1>3 |
| SBP | (mm Hg) | 115.4 ± 2.8 | 114.2 ± 2.3 | 119.5 ± 2.2 | ns |
| MAP | (mm Hg) | 88.8 ± 1.9 | 86.6 ± 1.6 | 88.1 ± 1.5 | ns |
| DBP | (mm Hg) | 73.5 ± 1.5 | 71.3 ± 1.3 | 71.5 ± 1.2 | ns |
| Heart rate | (bpm) | 67.4 ± 1.7 | 65.4 ± 1.4 | 61.4 ± 1.3 | * 1 = 2>3 |
| **Vascular Function** | | | | | |
| FMD | (%) | 7.0 ± 0.6 | 6.7 ± 0.5 | 7.5 ± 0.5 | ns |
| cAIx[d] | (%) | 19.3 ± 2.3 | 17.1 ± 2.2 | 18.3 ± 1.8 | ns |
| cfPWV[d] | (m/s) | 6.8 ± 0.2 | 6.5 ± 0.2 | 6.5 ± 0.2 | ns |
| crPWV[d] | (m/s) | 7.5 ± 0.3 | 8.2 ± 0.2 | 7.5 ± 0.2 | [t] 2>3 = 1 |
| cPP[cd] | (mm Hg) | 29.1 ± 1.7 | 28.3 ± 1.6 | 34.5 ± 1.3 | * 2 = 1<3 |
| Thigh Vmax50[e] | (ml) | 3.3 ± 0.2 | 3.0 ± 0.2 | 3.7 ± 0.2 | * 2<3 |
| Calf Vmax50[e] | (ml) | 1.9 ± 0.1 | 1.9 ± 0.1 | 2.0 ± 0.1 | ns |
| EPC-CFU[c] | (U) | 4.2 ± 1.5 | 4.7 ± 2.1 | 4.9 ± 0.6 | ns |

abbreviations: see Table 3.

[a] adjusted means ± SE values displayed are from non-transformed variables.

[b] $\chi^2$ or ANCOVA of group differences;

*p < .05;

**p < .01;

***p < .001;

[t] p < .10; ns = not significant.

[c] group comparison reflects analysis of transformed values.

[d] ABC+ n = 24, ABC- n = 26, HIV- n = 43.

[e] ABC+ n = 24, ABC- n = 37, HIV- n = 46.

vascular endothelial dysfunction, arterial stiffness, and CVD pathogenesis [47]. The present study examined the association of ABC treatment in PWH with established early prognostic indices of vascular endothelial function, arterial stiffness and endothelial repair. The PWH groups were on stable ART regimens and were well-matched with each other, as well as with the HIV seronegative group. The major findings from the analyses of the final model were as follows: **1)** no group differences were observed in BMI, central adiposity, indices of metabolic regulation, CRP, or the ASCVD 10-year risk score; **2)** the ABC- group displayed greater dyslipidemia than the other groups; **3)** the vascular function analyses showed that the HIV+ groups displayed higher heart rate and lower cPP. Notably, the FMD measure of endothelial-

**Table 5. Subclinical cardiometabolic measures comparing study groups when controlling for demographic variables (sex, age, education) and prescribed medications (antihypertensive, statin-fibrate, antipsychotic, thyroid).**

| Measures | | ABC+ (1) | ABC- (2) | HIV- (3) | Comparison[a] |
|---|---|---|---|---|---|
| | | Mean SE | Mean SE | Mean SE | |
| **CVD Risk** | | | | | |
| BMI | (kg/m$^2$) | 28.6 ± 0.9 | 27.2 ± 0.7 | 28.0 ± 0.7 | ns |
| Waist circumference | (cm) | 95.5 ± 3.1 | 94.8 ± 2.4 | 94.0 ± 2.2 | ns |
| HOMA-IR[c] | (mass units) | 3.1 ± 0.4 | 2.5 ± 0.3 | 2.3 ± 0.3 | ns |
| FPG[c] | (mg/dl) | 93.4 ± 2.4 | 90.7 ± 1.9 | 89.3 ± 1.7 | ns |
| FPI[c] | (μU/ml) | 13.1 ± 1.4 | 10.8 ± 1.1 | 10.3 ± 1.0 | ns |
| CRP | (mg/L) | 2.4 ± 0.6 | 2.7 ± 0.5 | 2.8 ± 0.5 | ns |
| TG[c] | (mg/dl) | 117.8 ± 16.7 | 143.0 ± 13.1 | 106.7 ± 11.9 | **2>3 |
| TC | (mg/dl) | 175.7 ± 7.4 | 202.1 ± 5.8 | 180.6 ± 5.3 | **2>3 = 1 |
| LDL | (mg/dl) | 101.9 ± 6.6 | 121.9 ± 5.2 | 102.9 ± 4.7 | * 2>3 = 1 |
| HDL | (mg/dl) | 50.2 ± 3.3 | 51.5 ± 2.6 | 56.4 ± 2.4 | ns |
| TC/HDL | | 3.7 ± 0.3 | 4.2 ± 0.2 | 3.4 ± 0.2 | **2>3 = 1 |
| TG/HDL[c] | | 2.6 ± 0.5 | 3.2 ± 0.4 | 2.2 ± 0.4 | **2>3 |
| SBP | (mm Hg) | 113.6 ± 3.0 | 115.2 ± 2.3 | 119.8 ± 2.1 | ns |
| MAP | (mm Hg) | 87.0 ± 2.1 | 87.5 ± 1.6 | 88.5 ± 1.5 | ns |
| DBP | (mm Hg) | 71.7 ± 1.7 | 72.2 ± 1.3 | 72.0 ± 1.2 | ns |
| Heart rate | (bpm) | 67.4 ± 1.9 | 65.4 ± 1.5 | 61.5 ± 1.4 | *1 = 2>3 |
| **Vascular Function** | | | | | |
| FMD | (%) | 7.6 ± 0.7 | 6.5 ± 0.5 | 7.4 ± 0.5 | ns |
| cAIx[d] | (%) | 18.4 ± 2.6 | 17.6 ± 2.3 | 18.6 ± 1.8 | ns |
| cfPWV[d] | (m/s) | 6.7 ± 0.2 | 6.6 ± 0.2 | 6.5 ± 0.2 | ns |
| crPWV[d] | (m/s) | 7.5 ± 0.3 | 8.2 ± 0.2 | 7.5 ± 0.2 | * 2>3 = 1 |
| cPP[cd] | (mm Hg) | 29.9 ± 1.9 | 28.6 ± 1.7 | 34.4 ± 1.3 | * 2 = 1<3 |
| Thigh Vmax50[e] | (ml) | 3.4 ± 0.2 | 3.0 ± 0.2 | 3.6 ± 0.2 | * 2<3 |
| Calf Vmax50[e] | (ml) | 2.0 ± 0.1 | 1.8 ± 0.1 | 2.0 ± 0.1 | ns |
| EPC-CFU[c] | (U) | 3.7 ± 1.5 | 4.9 ± 2.1 | 5.0 ± 0.6 | ns |

abbreviations: see Table 3.

[a] means ± SE values displayed are from non-transformed variables.

[b] $\chi^2$ or ANCOVA of group differences;

*p < .05;

**p < .01;

***p < .001;

[t]p < .10; ns = not significant.

[c] group comparison reflects analysis of transformed values.

[d] ABC+ n = 24, ABC- n = 26, HIV- n = 43.

[e] ABC+ n = 27, ABC- n = 38, HIV- n = 43.

dependent vasodilatory function, the EPC-CFU measure of endothelial repair, and specific indices of arterial stiffness (i.e., cAIx, cfPWV and calf Vmax50) did not differ among groups; **4)** when HIV related factors were controlled, no additional differences between the ABC+ and ABC- groups emerged; **5)** measures of crPWV and thigh Vmax50 indicated that the ABC-group compared with counterparts had greater arterial stiffness in these sites; and **6)** the EPC-CFU measure of vascular endothelial repair tended to be lower in the HIV+ groups than the HIV- group. Therefore, the present study found no substantial group differences that would suggest ABC treatment robustly enhances subclinical vascular pathogenesis.

**Table 6. Subclinical cardiometabolic measures comparing study groups when controlling for demographic variables (sex, age, education) and prescribed medications (antihypertensive, statin-fibrate, antipsychotic, thyroid) and substance abuse score.**

| Measures | | ABC+ (1) | ABC- (2) | HIV- (3) | Comparison[a] |
|---|---|---|---|---|---|
| | | Mean SE | Mean SE | Mean SE | |
| **CVD Risk** | | | | | |
| BMI | (kg/m$^2$) | 28.4 ± 1.0 | 27.4 ± 0.8 | 27.9 ± 0.7 | ns |
| Waist circumference | (cm) | 94.7 ± 3.2 | 94.8 ± 2.4 | 94.0 ± 2.2 | ns |
| HOMA-IR[c] | (mass units) | 3.0 ± 0.4 | 2.6 ± 0.3 | 2.3 ± 0.3 | ns |
| FPG[c] | (mg/dl) | 92.9 ± 2.4 | 91.2 ± 1.9 | 89.1 ± 1.7 | ns |
| FPI[c] | (μU/ml) | 12.9 ± 1.4 | 11.0 ± 1.1 | 10.2 ± 1.0 | ns |
| CRP | (mg/L) | 2.4 ± 0.6 | 2.7 ± 0.5 | 2.8 ± 0.5 | ns |
| TG[c] | (mg/dl) | 118.8 ± 17.0 | 141.8 ± 13.5 | 107.0 ± 12.0 | * 2>3 |
| TC | (mg/dl) | 173.5 ± 7.5 | 204.6 ± 5.9 | 179.8 ± 5.3 | **2>3 = 1 |
| LDL | (mg/dl) | 99.6 ± 6.6 | 124.6 ± 5.2 | 102.1 ± 4.7 | **2>3 = 1 |
| HDL | (mg/dl) | 50.2 ± 3.4 | 51.5 ± 2.7 | 56.3 ± 2.4 | ns |
| TC/HDL | | 3.6 ± 0.3 | 4.3 ± 0.2 | 3.4 ± 0.2 | **2>3 = 1 |
| TG/HDL[c] | | 2.6 ± 0.5 | 3.2 ± 0.4 | 2.2 ± 0.4 | **2>3 |
| SBP | (mm Hg) | 113.3 ± 3.0 | 115.5 ± 2.3 | 119.7 ± 2.1 | ns |
| MAP | (mm Hg) | 87.4 ± 2.1 | 87.1 ± 1.7 | 88.6 ± 1.5 | ns |
| DBP | (mm Hg) | 72.0 ± 1.7 | 71.9 ± 1.3 | 72.1 ± 1.2 | ns |
| Heart rate | (bpm) | 67.4 ± 1.9 | 65.4 ± 1.5 | 61.5 ± 1.4 | * 1 = 2>3 |
| **Vascular Function** | | | | | |
| FMD | (%) | 7.5 ± 0.7 | 6.6 ± 0.5 | 7.4 ± 0.5 | ns |
| cAIx[d] | (%) | 19.2 ± 2.6 | 16.7 ± 2.4 | 18.7 ± 1.8 | ns |
| cfPWV[d] | (m/s) | 6.7 ± 0.3 | 6.6 ± 0.2 | 6.5 ± 0.2 | ns |
| crPWV[d] | (m/s) | 7.6 ± 0.3 | 8.2 ± 0.2 | 7.5 ± 0.2 | t 2>3 |
| cPP[cd] | (mm Hg) | 28.6 ± 2.0 | 28.8 ± 1.8 | 34.4 ± 1.3 | * 2 = 1<3 |
| Thigh Vmax50[e] | (ml) | 3.4 ± 0.2 | 2.9 ± 0.2 | 3.6 ± 0.2 | * 2<3 |
| Calf Vmax50[e] | (ml) | 2.0 ± 0.1 | 1.8 ± 0.1 | 2.0 ± 0.1 | ns |
| EPC-CFU[c] | (U) | 4.1 ± 1.5 | 4.4 ± 2.1 | 5.2 ± 0.6 | t 1 = 2<3 |

abbreviations: see Table 3.

[a] means ± SE values displayed are from non-transformed variables.

[b] $\chi^2$ or ANCOVA of group differences;

*p < .05;

**p < .01;

***p < .001;

[t] p < .10; ns = not significant.

[c] group comparison reflects analysis of transformed values.

[d] ABC+ n = 24, ABC- n = 26, HIV- n = 43.

[e] ABC+ n = 27, ABC- n = 38, HIV- n = 43.

Previously, a channeling or selection bias has been identified in cohort studies, wherein PWH participants who were prescribed ABC evidenced more elevated CVD-related morbidity [21]. Of note, a recent meta-analysis indicated that since 2008 the prescription of ABC declined in PWH, particularly among persons with moderate to severe CVD risk [20]. Thus, since this time a reverse channeling bias may be present in ART prescription practice. The present study design employed a group recruitment procedure that matched comparison groups with the ABC+ group on sex, ethnicity/race and BMI. We reasoned that this procedure

together with strict study eligibility criteria would counter the previously reported channeling or selection bias in CVD risk attributed to persons treated with ABC. Eligibility criteria were employed to exclude persons with diagnosed conditions, wherein treatment may have cardio-metabolic influences. However, persons with hypertension and dyslipidemia were permitted into the study. As can be seen in Table 1, the ABC-treated group compared with other groups were comprised of more persons treated for hypertension and dyslipidemia. Statistical control was implemented where necessary to adjust for demographic variables, medications for comorbid conditions, and substance abuse history. Groups evidenced no differences on the group matching variables, key indices of traditional CVD risk (other than non-HDL choles-terol levels and triglyceridemia), or substance dependence and abuse classification. Thus, the PWH groups were well-matched with each other, and with the HIV- group.

Virus-related mechanisms possibly linked to proinflammatory effects of HIV proteins due to viral replication and CD4 depletion have been posited to account for a proportion of ele-vated CVD risk in PWH [48, 49]. Of note, elevated inflammatory levels are observed in PWH even when their viral loads are suppressed and CD4 counts preserved [50]. Simmering low-grade inflammation, viral reactivation of comorbid chronic infections, gut microbial transloca-tion, and immune-senescence are putative underlying factors associated with HIV spectrum disease and advanced aging [50]. In the present study, the HIV+ groups did not differ on ART duration and were largely comprised of persons with undetectable viral load. In addition, anal-yses indicated no group differences in CRP levels. When analyses controlled for HIV disease severity, indexed by HIV viral load and CD4 count, the observed lack of differences in vascular outcomes between the ABC-treated and -nontreated groups suggest that the vascular out-comes were independent of these HIV-related factors. Of note, the HIV+ groups displayed lower cPP and higher HR than other groups, albeit well within normative ranges. These differ-ences suggest that the HIV+ participants may have more diminished stroke volume that is compensated partially but incompletely by HR. Because of the association of HIV infection with cardiomyopathy, it is possible that the observed hemodynamic alterations reflect adverse preclinical cardiac remodeling resulting in changes in cardiac structure and function [51]. This hypothesis is supported by previous studies that have documented elevated left ventricu-lar mass and diastolic dysfunction in PWH compared with seronegative counterparts [52, 53].

It is notable that other studies have used similar vascular function measures (FMD and arte-rial stiffness) as used in the present study to clarify the association of ABC treatment with sub-clinical CVD risk in PWH [22–24]. In these previous reports, ABC treatment was associated with impaired vascular endothelial dilation and elevated arterial stiffness. Indeed, an in vitro study of the effect of ABC on human endothelial cells indicated that drug exposure resulted in diminished endothelial nitric oxide synthase and increased oxidative stress, supporting the notion that ABC treatment may be deleterious to the vascular endothelium [25]. However, other studies have failed to find an association of ABC treatment with vascular endothelial dys-function indexed by FMD or carotid intimal wall thickness [24, 26, 27]. The latter studies sup-port recent meta-analyses of randomized clinical trials, wherein an association of ABC use with myocardial infarction risk was not found [54–57]. Similarly, other studies have shown that ABC treatment is not associated with insulin resistance, dyslipidemia, central adiposity or other traditional CVD risk factors [58, 59]. The present findings are consistent with these stud-ies and have extended these findings to measures examining arterial stiffness using different methodologies assessing both central and peripheral sites and endothelial repair function. Of note, study findings were independent of demographics, medication use prescribed for comor-bid conditions, and substance abuse.

The present analyses also revealed that the ABC- group displayed some evidence of greater stiffness in the arterial supply to the radial and thigh circulations. Peripheral arterial stiffness is

associated with CVD, and diminished compliance of the arterial circulation in the thigh is a particularly strong indicator of CVD risk [60–62]. It is unclear why these signs of CVD risk were distinguished in the ABC- group. This difference is supported by the trend toward more diminished vascular endothelial repair in the HIV+ groups compared with the seronegative group, independent of control variables. This finding suggests that underlying mechanisms of HIV infection may be inhibiting EPC reparatory function. No previous study has examined this index of endothelial repair in the context of an evaluation of ABC treatment. It should be noted that, although EPC-CFU is indicative of endothelial repair function, it does not necessarily imply active participation of EPCs in endothelial repair. In PWH, diminished circulating EPC trafficking has been identified [45, 63]. A recent study has shown that low levels of circulating EPC cells were also associated with CVD risk factor burden in older HIV+ men [64]. Further research aimed at determining whether endothelial repair function is mechanistically related to the observed arterial stiffening at these circulatory sites in PWH may be warranted.

## Limitations

There are several limiting issues to consider. The cross-sectional design has many issues, wherein the primary limitation is that causal attribution is not possible. Nevertheless, the study provided a comprehensive examination of prognostic indices of early vascular pathophysiology. Although the study participants spanned several HIV exposure categories and manifested stages of infection progression, the present cohort was comprised mostly of low socioeconomic status Hispanic men. Notably, however, the cohort characteristics were representative of the local community. By strategic design, the study did not enroll persons diagnosed with CVD, whereby the impact of more advanced disease process could be examined. Although the study recruitment procedure matched ABC- and HIV- participants to ABC+ participants by the sex, ethnicity/race and BMI of the ABC+ group, groups were not matched on age. Findings indicated that HIV- subjects were younger on average than the HIV+ groups. However, all analyses controlled for age. Hence, study findings were independent of any differences due to age. We have argued that the study procedures using group matching, enrollment restriction and covariate analysis techniques have diminished possible effects of channeling or selection bias inherent in ABC-treatment. Nevertheless, it is difficult to maintain that no residual confounding of factors not considered in the study design remained. In addition, even after adjusting for potential confounding variables, there may still be some selection or other bias that cannot be corrected analytically. Although no robust evidence of elevated subclinical CVD was observed in ABC-treated PWH, the present cohort was younger (30-50 yrs) than most cohorts in the aforementioned studies assessing the association of ABC with vascular indices. Thus, it is possible that more advanced aging may be mediating the diminished vascular function observed in previous ABC studies [e.g., 22]. In this study, the treatment duration of the current ART regimen with or without ABC was about 3 years in PWH. However, it is possible that the deleterious effect of ABC treatment on vascular endothelial function, arterial stiffness and CVD risk requires more prolonged ABC exposure and evaluation of medication interactions [65]. One of the problems inherent in evaluating ART regimen effects in cross-sectional studies is that obtaining medical records to account for prior ART regimen history is difficult and often not possible. Even were this information obtained, large sample sizes would be required to overcome ART treatment variability due to changes in medication prescription, treatment duration and the evolution of ART pharmaceutical development and clinical application. To overcome such study design issues, longitudinal studies of treatment-naïve PWH could be undertaken, and have, but the exposure duration required to observe even subclinical CVD outcomes may be impractical for logistic and clinical reasons.

## Conclusion

When PWH on stable ART with or without ABC in the regimen were well-matched and compared with HIV seronegative individuals, no substantial group differences in indices of traditional cardiovascular disease risk, or in vascular endothelial dilatory function, arterial stiffness and endothelial repair were observed. These findings were independent of demographics, medication use and substance abuse. Considering that the employed indices of cardiometabolic and vascular function are considered strong prognostic indicators of CVD risk, our study provides no robust evidence that ABC is unsafe. Study conclusions are limited to the characteristics of the cohort, which were representative of the local community, but younger and of lower CVD risk than comparable studies of the impact of ABC treatment. Moreover, study conclusions must be tempered because the temporal complexities of HIV spectrum disease and related symptoms, comorbidities and medical treatment, as well as ART prescription history and medication interactions could not be considered in the context of this study design. Although present findings indicate that vascular endothelial dysfunction and arterial stiffening are not likely mechanisms by which ABC deleteriously influences subclinical CVD risk in PWH, it is possible that ABC treatment acts synergistically with aging, HIV disease progression, immunosenescence and treatment duration. Further research delineating the interplay of these factors may be of relevance in assessing the relationship of ABC treatment and CVD events.

## Supporting information

**S1 Data.**
(XLSX)

## Acknowledgments

We are grateful to Barbara Lang, Alejandro Chedebeau, Blaire Hall, Rosa Hernandez, Valeria Porras, Solorzano Rubio, Dalhila and the Miami CFAR Clinical and behavioral core team.

## Author Contributions

**Conceptualization:** Claudia A. Martinez, Michael A. Kolber, Ivonne H. Schulman, Barry E. Hurwitz.

**Data curation:** Claudia A. Martinez, Mollie S. Pester, Alex Gonzalez, Maria L. Alcaide, Barry E. Hurwitz.

**Formal analysis:** Alex Gonzalez, Michaela Larson, Barry E. Hurwitz.

**Funding acquisition:** Claudia A. Martinez.

**Investigation:** Claudia A. Martinez, Meela Parker, Armando Mendez, Ivonne H. Schulman, Maria L. Alcaide, Barry E. Hurwitz.

**Methodology:** Claudia A. Martinez, Michael A. Kolber, Ivonne H. Schulman, Barry E. Hurwitz.

**Project administration:** Claudia A. Martinez, Barry E. Hurwitz.

**Resources:** Armando Mendez, Jeffrey K. Raines, Maria L. Alcaide, Barry E. Hurwitz.

**Software:** Jeffrey K. Raines, Barry E. Hurwitz.

**Supervision:** Claudia A. Martinez, Michael A. Kolber, Barry E. Hurwitz.

**Writing – original draft:** Claudia A. Martinez, Barry E. Hurwitz.

**Writing – review & editing:** Claudia A. Martinez, Rishi Rikhi, Michaela Larson, Jennifer Chavez, Ivonne H. Schulman, Maria L. Alcaide, Barry E. Hurwitz.

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
