## [Decision Letter · Decision Letter 0]

30 Oct 2021

PONE-D-21-20487Abacavir Antiretroviral Therapy

and Indices of Subclinical Vascular Disease in Persons with HIVPLOS ONE

Dear Dr. Martinez,

Thank you for submitting your manuscript to PLOS ONE. After careful consideration, we feel that it has merit but does not fully meet PLOS ONE’s publication criteria as it currently stands. Therefore, we invite you to submit a revised version of the manuscript that addresses the points raised during the review process.

We look forward to receiving your revised manuscript.

Kind regards,

Huimin Yan

Academic Editor

PLOS ONE

Journal Requirements:

Reviewers' comments:

Reviewer's Responses to Questions

**Comments to the Author**

1. Is the manuscript technically sound, and do the data support the conclusions?

Reviewer #1: Yes

Reviewer #2: Yes

2. Has the statistical analysis been performed appropriately and rigorously? 

Reviewer #1: Yes

Reviewer #2: Yes

3. Have the authors made all data underlying the findings in their manuscript fully available?

Reviewer #1: Yes

Reviewer #2: Yes

4. Is the manuscript presented in an intelligible fashion and written in standard English?

Reviewer #1: Yes

Reviewer #2: Yes

5. Review Comments to the Author

Reviewer #1: The present study assess several pathways of vascular function in PWH on an stable ART including or not ABC. A very extensive assessment of vascular function has been carried out in 30 to 50 years of age participants. The ABC + group did not present any increase in the vascular indices assessed, compared with and ABC – and HIV – groups, even when an ANCOVA analyses was performed.

Minor comments:

Could the authors explain why persons with didanosine or women postmenopausal were excluded.

A limitation of the study is that participants have not been matched by age a factor closely related with cardiovascular risk. I suggest to add this issue in the limitations.

Could the authors define mildly and moderately increase of TG and TC. I suggest to use standard definitions: TG >150 mg/d Land TC>200 mg/dL.

Reviewer #2: This manuscript describes no differences in the ABC-based group vs. non-ABC treated PLWH regarding FMD, arterial stiffness, or plasma endothelial repair cells.

As an observational study, it may be biased due to its non-randomized nature. The authors have made several reasonable adjustments to overcome this potential bias, and they found no differences between groups on the variables mentioned earlier. The methods are adequately described, and statistical analyses are detailed and rigorous.

The association of CVD risk and abacavir use is a matter of open controversy. Basal CVD risk may be a critical factor in that ABC would be associated with an increased CVD risk in patients with established cardiovascular disease; in this scenario, ABC could induce platelet activation. Unfortunately, the findings from observational studies won’t be the definitive answer for this complex clinical problem, especially in patients with a low CVD risk as the study patients should have, with a mean age of 42 years. Therefore, authors must present the estimated CVD risk on some scale, for example, ASCVD.

The conclusions of the abstract should indicate the characteristics of the population studied. Also, it is not possible either to affirm with their results that ABC is not associated with increased CVD risk, which is a clinical variable that they have not analyzed. Instead, it is not associated with changes in FMD or rigidity or the number of EPC.

Minor comment: slightly elevated triglycerides on table 1; from a clinical point of view, this may be easy to understand, but I suggest displaying the triglyceride values with their median and IQR.

6. PLOS authors have the option to publish the peer review history of their article (what does this mean?). If published, this will include your full peer review and any attached files.

Reviewer #1: No

Reviewer #2: **Yes: **Vicente Estrada

---

## [Author Response · Author response to Decision Letter 0]

24 Dec 2021

Reviewer #1: 

1. Could the authors explain why persons with didanosine or women postmenopausal were excluded.

Response: an explanation has been inserted; see pg 3 para 1 lines 4-8

2. A limitation of the study is that participants have not been matched by age a factor closely related with cardiovascular risk. I suggest to add this issue in the limitations.

Response: comment has been added regarding this limitation; see pg 8 para 3 lines 8-12

3. Could the authors define mildly and moderately increase of TG and TC. I suggest to use standard definitions: TG >150 mg/d Land TC>200 mg/dL.

Response: we have added these criteria in the footnotes of Table 1

Reviewer #2:

1. …the authors must present the estimated CVD risk on some scale, for example, ASCVD

Response: Thank you for raising this issue. As suggested the ASCVD 10-year risk score has been reported in the results, Table 1, and included in the discussion.

2. The conclusions of the abstract should indicate the characteristics of the population studied. 

Response: as suggested; see Abstract line 19

3. It is not possible either to affirm with their results that ABC is not associated with increased CVD risk, which is a clinical variable that they have not analyzed. Instead, it is not associated with changes in FMD or rigidity or the number of EPC. 

Response: as suggested ASCVD parameters have been inserted in results, table 1, and discussion.

4. Minor comment: slightly elevated triglycerides on table 1; from a clinical point of view, this may be easy to understand, but I suggest displaying the triglyceride values with their median and IQR.

Response: median and IQR values have been inserted for triglycerides and total cholesterol; see Table 1

---

## [Decision Letter · Decision Letter 1]

11 Feb 2022

Abacavir Antiretroviral Therapy

and Indices of Subclinical Vascular Disease in Persons with HIV

PONE-D-21-20487R1

Dear Dr. Martinez,

We’re pleased to inform you that your manuscript has been judged scientifically suitable for publication and will be formally accepted for publication once it meets all outstanding technical requirements.

Kind regards,

Huimin Yan

Academic Editor

PLOS ONE

Additional Editor Comments (optional):

Reviewers' comments:

Reviewer's Responses to Questions

**Comments to the Author**

1. If the authors have adequately addressed your comments raised in a previous round of review and you feel that this manuscript is now acceptable for publication, you may indicate that here to bypass the “Comments to the Author” section, enter your conflict of interest statement in the “Confidential to Editor” section, and submit your "Accept" recommendation.

Reviewer #1: All comments have been addressed

Reviewer #2: All comments have been addressed

2. Is the manuscript technically sound, and do the data support the conclusions?

Reviewer #1: Yes

Reviewer #2: Yes

3. Has the statistical analysis been performed appropriately and rigorously? 

Reviewer #1: Yes

Reviewer #2: Yes

4. Have the authors made all data underlying the findings in their manuscript fully available?

Reviewer #1: Yes

Reviewer #2: Yes

5. Is the manuscript presented in an intelligible fashion and written in standard English?

Reviewer #1: Yes

Reviewer #2: Yes

6. Review Comments to the Author

Reviewer #1: (No Response)

Reviewer #2: (No Response)

7. PLOS authors have the option to publish the peer review history of their article (what does this mean?). If published, this will include your full peer review and any attached files.

Reviewer #1: No

Reviewer #2: **Yes: **Vicente Estrada

---

## [Editor Report · Acceptance letter]

28 Feb 2022

PONE-D-21-20487R1 

Abacavir Antiretroviral Therapy
and Indices of Subclinical Vascular Disease in Persons with HIV 

Dear Dr. Martinez:

I'm pleased to inform you that your manuscript has been deemed suitable for publication in PLOS ONE. Congratulations! Your manuscript is now with our production department. 

Kind regards, 

on behalf of

Dr. Huimin Yan 

Academic Editor

PLOS ONE